# Essential function of the alveolin network in the subpellicular microtubules and conoid assembly in *Toxoplasma gondii*

Nicolò Tosetti[1], Nicolas Dos Santos Pacheco[1], Eloïse Bertiaux[2], Bohumil Maco[1], Lorène Bournonville[2], Virginie Hamel[2], Paul Guichard[2], Dominique Soldati-Favre[1]*

[1]Department of Microbiology and Molecular Medicine, Faculty of Medicine, University of Geneva, Geneva, Switzerland; [2]Department of Cell Biology, Sciences III, University of Geneva, Geneva, Switzerland

**Abstract** The coccidian subgroup of Apicomplexa possesses an apical complex harboring a conoid, made of unique tubulin polymer fibers. This enigmatic organelle extrudes in extracellular invasive parasites and is associated to the apical polar ring (APR). The APR serves as microtubule-organizing center for the 22 subpellicular microtubules (SPMTs) that are linked to a patchwork of flattened vesicles, via an intricate network composed of alveolins. Here, we capitalize on ultrastructure expansion microscopy (U-ExM) to localize the *Toxoplasma gondii* Apical Cap protein 9 (AC9) and its partner AC10, identified by BioID, to the alveolin network and intercalated between the SPMTs. Parasites conditionally depleted in AC9 or AC10 replicate normally but are defective in microneme secretion and fail to invade and egress from infected cells. Electron microscopy revealed that the mature parasite mutants are conoidless, while U-ExM highlighted the disorganization of the SPMTs which likely results in the catastrophic loss of APR and conoid.

## Introduction

*Toxoplasma gondii* belongs to the phylum of Apicomplexa that groups numerous parasitic protozoans causing severe diseases in humans and animals. As part of the superphylum of Alveolata, the Apicomplexa are characterized by the presence of the alveoli, which consist in small flattened single-membrane sacs, underlying the plasma membrane (PM) to form the inner membrane complex (IMC) of the parasite. A rigid alveolin network composed of intermediate filament-like proteins is lining the cytoplasmic side of the IMC. The alveolin network is made of proteins sharing conserved repeat motifs called alveolins, which together with the IMC span the length of the parasite from the apical polar ring (APR) to the basal complex (*Mann and Beckers, 2001*; *Gould et al., 2008*; *Anderson-White et al., 2011*). The IMC plays an essential role in parasite motility by anchoring the actomyosin system (*Frénal et al., 2017*) and serves as structural scaffold during daughter cells formation within the mother cell, an asexual form of reproduction referred to as endodyogeny (*Francia and Striepen, 2014*).

The IMC is arranged in a series of rectangular plates sutured together. Several IMC sutures components (ISCs) have been localized to the transversal and longitudinal sutures between alveolar sacs (*Chen et al., 2015*; *Chen et al., 2017*). The parasite is capped by a single cone-shaped plate called the apical cap. Several proteins have been reported to localize at the apical cap including the IMC sub-compartment protein 1 (ISP1) (*Beck et al., 2010*) and nine apical cap proteins (called AC1 to AC9) (*Chen et al., 2015*; *Chen et al., 2017*). Centrin2 labels a peripheral ring of six annuli at the boundary of the apical plate and the rest of the alveolar plates (*Hu et al., 2006*). Beneath the alveolin network, a set of 22 subpellicular microtubules (SPMT) spanning two third of the parasite length confers the elongated shape to the tachyzoites. The SPTMs originate at the APR, which serves as

*For correspondence:
Dominique.Soldati-Favre@unige.ch

microtubule-organizing center (MTOC). Ultrastructural studies have highlighted tight connections between the IMC complex and the SPMTs. More specifically, freeze-fractured studies showed double rows of inner membranous particles (IMPs) at the IMC surface arranged in spiraling longitudinal rows reminiscent of SPMT path exhibiting a repeating pattern with a 32 nm periodicity. This 32 nm periodicity was also observed in the single row of IMP associated with SPMTs (*Morrissette et al., 1997*).

The APR and the secretory organelles, rhoptries (implicated in invasion) and micronemes (implicated in motility, invasion and egress), are common features of the apical complex conserved in all motile and invasive apicomplexans. Members of the coccidian subgroup of the phylum, including *Toxoplasma, Neospora, Besniotia, Cyclospora, Sarcocystis* and *Eimeria*, possess an additional organelle termed the conoid. This dynamic organelle is formed by 14 atypical comma-shape tubulin fibers composed of ~9 protofilaments differing from classical microtubules in that they do not form a closed tubule but rather an open comma shaped structure (*Hu et al., 2002*). The conoid is enclosed between the APR and two preconoidal rings (PCRs) at the apex of the parasite. In addition, two short intraconoidal microtubules are present within the conoid (*Morrissette, 2015*). During division, the conoid is retracted posterior to the APR whereas it protrudes through the APR in a calcium-dependent manner in motile parasites (*Mondragon and Frixione, 1996*). The calcium-dependent protein kinase 1 (CDPK1) is an essential regulator of microneme exocytosis (*Lourido et al., 2010*) and its deletion results in inhibition of conoid protrusion and a block of the apico-basal flux of F-actin, essential for parasite motility (*Tosetti et al., 2019*). The mechanistic contribution of the conoid in motility and invasion has not been elucidated to date. During evolution, the conoid was lost in Haematozoa which include *Plasmodium* spp. and piroplasmidia (*Theileria* and *Babesia*). In contrast, the conoid is still present in deeply branching *Cryptosporidium* spp. and *Gregarina* spp. Other more distantly related Alveolata (colpodellids, perkinsids and chromerids) possess a similar structure referred to as incomplete conoid or pseudoconoid, built from apical MTs but lacking the APR, which suggests that early branching apicomplexans harbored an ancestral apical complex along with secretory organelles (*Gubbels and Duraisingh, 2012*). Intriguingly, the SAS6-like protein, which is localized near the base of the flagellum of *Trypanosoma brucei*, is found at the preconoidal rings of *T. gondii* tachyzoites that do not possess a flagellum, raising questions about the origin of the conoid (*de Leon et al., 2013*). SAS6-like has also been localized apically in both ookinetes and sporozoites but absent in merozoites of *Plasmodium berghei*, revealing that the morphology and composition of the apical complex might vary between different parasitic life stages and yet retained conserved proteins (*Wall et al., 2016*).

More than 250 proteins were enriched in the conoid fraction by a proteomic approach (*Hu et al., 2006*) but only a few have been functionally investigated. Knockdown of the APR protein RNG2 results in microneme secretion defect (*Katris et al., 2014*), while the conoid-associated MyoH and the likely associated calmodulin-like proteins are all required for gliding motility that powers invasion and egress (*Graindorge et al., 2016*; *Long et al., 2017a*). Additionally, the apical complex methyltransferase (AKMT) (*Heaslip et al., 2011*), the formin 1 (FRM1) (*Tosetti et al., 2019*) and the glideosome-associated connector (GAC) (*Jacot et al., 2016*) are apical proteins conserved across the Apicomplexa that play essential roles in parasite motility, invasion and egress (*Frénal et al., 2017*). Among the proteins involved in stability of the apical complex, loss of microtubules binding doublecortin (DCX)-domain protein results in shorter disordered conoid with subsequent defect in invasion (*Nagayasu et al., 2017*). DCX was later shown to bundle the conoidal tubulin fibers into comma-shape structures likely conferring the cone-shaped configuration of the conoid (*Leung et al., 2020*). Moreover, the double knockout of kinesin A and apical polar ring protein 1 (APR1) caused a fragmentation of the APR with a conoid frequently partially detached resulting in impairment of microneme secretion (*Leung et al., 2017*). Lastly, conoid protein hub 1 (CPH1) is a conserved apicomplexan protein essential for parasite motility, invasion and egress that regulates conoid stability in extracellular parasites without impacting on microneme secretion (*Long et al., 2017b*). Similarly, a recently described protein, the MAP kinase ERK7, localizing to the apical cap is reported to be involved in conoid assembly without impacting on microneme secretion (*O'Shaughnessy et al., 2020*). Here, we have used ultrastructure expansion microscopy (U-ExM) to precisely localize AC9 and its partner AC10, identified by BioID, to the alveolin network but exclusively in the apical cap region in-between the SPMTs. Conditional depletion of AC9 and AC10 established that the two proteins are essential for parasite motility invasion and egress, while parasite replication is not affected.

U-ExM and electron microscopy revealed striking morphological defects. AC9 or AC10 depleted mature parasites are conoidless, devoid of APR and their SPMTs are disorganized. The arrangement of AC9 and AC10 suggests that both proteins act as a glue between SPMTs, a prerequisite to maintain the overall architecture of the key components of the apical complex when the parasite matures.

## Results

### AC10 is a new apical cap protein closely associated to AC9

T. gondii AC9 is an hypothetical protein exhibiting no informative protein domain to unravel its function but a strong negative fitness score indicative of essentiality (*Sidik et al., 2016*). AC9 was modified by CRISPR/Cas9 editing at the endogenous locus to generate a fusion with the mini auxin-induced degron (mAID) and HA-tag at the C-terminus to localize and rapidly downregulate the protein upon addition of auxin (IAA) (*Brown et al., 2017*; *Figure 1—figure supplement 1A*). As previously reported (*Chen et al., 2017*), AC9 colocalizes with ISP1 at the most apical plate of the IMC termed apical cap, delimited by the peripheral annuli stained by centrin2 (Cen2). AC9 appears very early during endodyogeny before the emergence of the daughter parasite scaffold stained with IMC1 antibodies (*Figure 1A* and *Figure 1—figure supplement 1B*). Further colocalization with other Ty-tagged proteins at their endogenous loci, revealed that AC9 surrounds the APR marked by RNG2-Ty (*Katris et al., 2014*) and the conoid localized with CPH1-Ty (*Figure 1B*). A second copy of the gliding associated protein 70 (GAP70)-mycGFP was introduced in the AC9-mAID-HA strain. Like GAP45, GAP70 is acylated and anchored at both the IMC and the PM spanning the space between those two membranes (*Frénal et al., 2010*). GAP70-mycGFP localized slightly more apically compared to AC9, which remains confined to the apical plate of the IMC (*Figure 1C*). Next, fractionation assay revealed that AC9 is completely resistant to extraction with the non-ionic detergent Triton X-100 but is solubilized in sodium carbonate at pH 11. This suggests that AC9 is an alveolin network resident protein associated to other alveolins proteins or a peripheral IMC protein (*Figure 1D*). Noteworthy, AC9-mAID-HA migrated on SDS-PAGE at around 70 kDa, slightly higher than the predicted size (60 kDa). To get insights into AC9 potential interacting partners, we opted for the proximity labeling by the biotin ligase BirA approach (*Long et al., 2018*) given the limited solubility of AC9. A mycBirA construct was inserted at the endogenous C-terminus locus of AC9 by CRISPR/Cas9 (*Figure 1—figure supplement 1C*) and shown to be correctly targeted to the apical cap (*Figure 1—figure supplement 1D*). To assess whether AC9-mycBirA was able to biotinylate proximal proteins, we performed IFA with fluorophore-conjugated streptavidin. In absence of biotin, apicoplast endogenously biotinylated proteins were observed while addition of biotin for 24 hr resulted in accumulation of biotinylated proteins in the apical cap, proving that the enzyme is active at the right position (*Figure 1—figure supplement 1D*). Of relevance, deoxycholate extracted parasites showed a robust streptavidin staining around the apical cortical microtubules just beneath the conoid indicating biotinylation of components of the parasite cytoskeleton (*Figure 1—figure supplement 1D*). Analysis of parasites total lysates by WB showed an enrichment of biotinylated proteins (*Figure 1—figure supplement 1E*). Immunoprecipitation with streptavidin magnetic beads was performed with parasites lysed in cytoskeletal buffer as previously described (*Long et al., 2018*; *Figure 1—figure supplement 1F*) followed by mass spectrometry analysis. Acetyl-CoA carboxylase ACC1 and pyruvate carboxylase are the endogenously biotinylated proteins typically found as top hit in wild type parasites. The clear top hit was a hypothetical protein (TGGT1_292950) followed by two apical cap proteins (AC2 and AC8) and IMC/cytoskeletal proteins (*Supplementary file 1* and *Supplementary file 2*). Candidates were further filtered by expression patterns similar to AC9 (*Figure 1—figure supplement 1G*). TGGT1_292950 exhibited a very similar cyclic mRNA expression pattern compared to AC9 (*Figure 1E*). Upon C-terminal Ty-epitope tagging, the product of the gene colocalized with AC9 in mature parasites and daughter cells and hence was named AC10 (*Figure 1F*). Like AC9, AC10 was showed to be partially solubilized by sodium carbonate and completely insoluble in Triton X-100 using the AC10-mAID-HA strain, indicating its alveolin network association (*Figure 1G*).

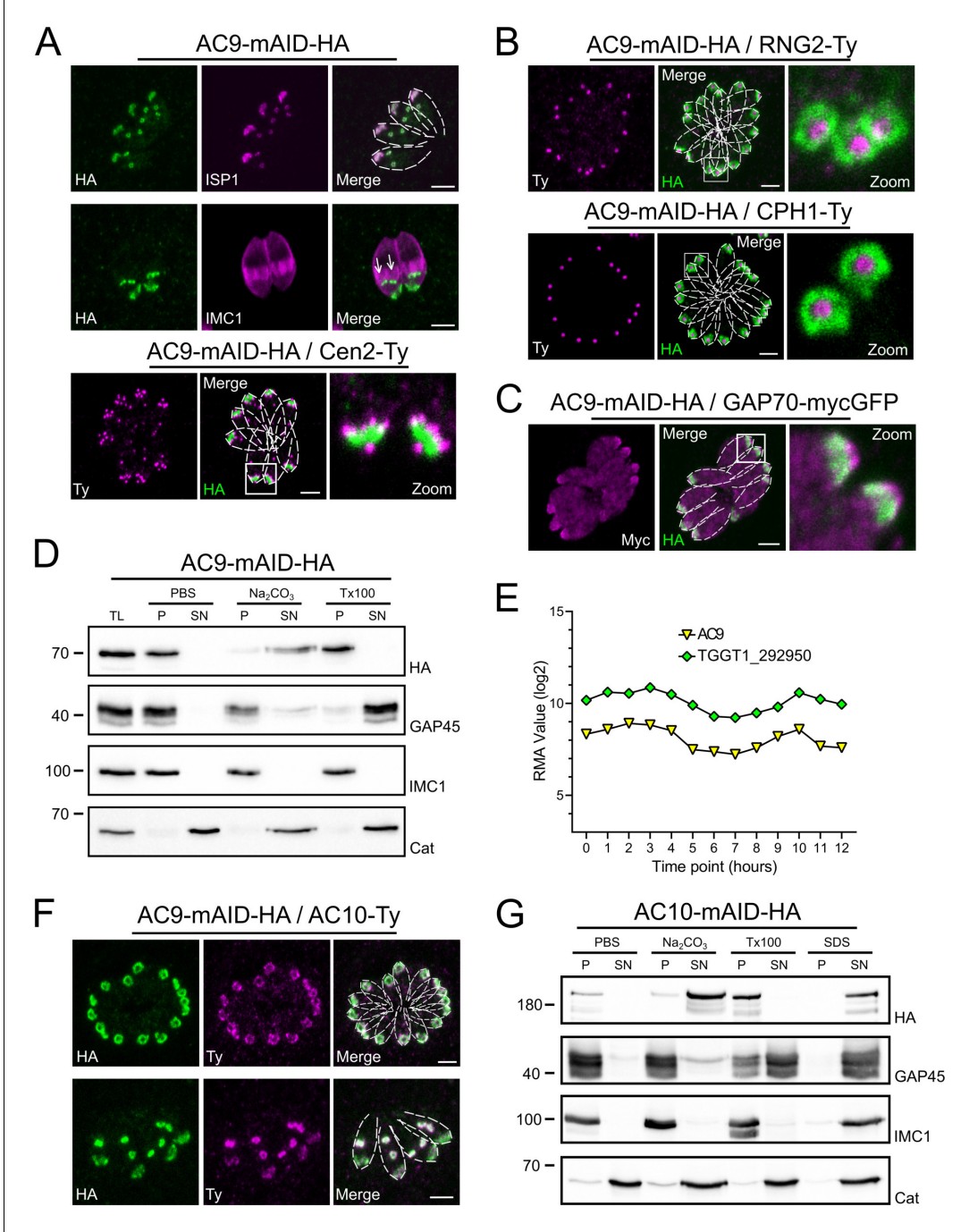

**Figure 1.** AC9 and the newly discovered AC10 are alveolin network proteins. (**A**) AC9-mAID-HA localized to the apical cap of both mature and daughter cells (arrows). AC9 is confined to the apical IMC plate delimited by the Centrin2 marker. (**B**) AC9 surrounds the conoid (CPH1 marker) and the apical polar ring (RNG2 marker). (**C**) Colocalization with GAP70 revealed that AC9 is likely confined in the IMC/subpellicular network (SPN) and not associated with the PM. (**D**) AC9-mAID-HA is partially solubilized by sodium carbonate and precipitates in Triton X-100 insoluble fraction. (**E**) TGGT1_292950 was the top biotinylated hit. This hypothetical protein displayed an identical cyclic expression pattern compared to AC9. (**F**) TGGT1_292950 colocalized with AC9 in both daughter and mature parasite and was subsequently named AC10. (**G**) Similar to AC9, AC10-mAID-HA is partially solubilized only by sodium carbonate. Scale bars = 2 μm.

The online version of this article includes the following figure supplement(s) for figure 1:

**Figure supplement 1.** Proximity biotinylation by AC9.

## AC9 and AC10 are recruited very early during daughter cells formation

ISP1 was localized to the apical cap along with several apical cap proteins (ACs) (*Chen et al., 2015*; *Chen et al., 2017*; *Beck et al., 2010*). Intriguingly, some ACs are recruited early during daughter cells formation while others appear later, highlighting the existence of a temporal and spatial hierarchy of IMC and alveolin network formation during parasite endodyogeny (*Chen et al., 2015*). Colocalization experiments revealed that AC9 and AC10 are recruited very early to the daughter cytoskeleton, prior to ISP1 incorporation (*Figure 2A–B*); in contrast, AC2 and AC8 are recruited later (*Figure 2—figure supplement 1A–B*), as schematized in *Figure 2C*. The three heavily biotinylated proteins AC2, AC8 and AC10, were endogenously tagged at the C-terminus in the AC9-mAID-HA strain to assess further the interaction with AC9. Deletion of AC9 had no impact on the localization of AC2, AC8, AC10 and ISP1 (*Figure 2D* and *Figure 2—figure supplement 1A–C*). Conversely, epitope-tagged AC9 in AC10-mAID-HA disappeared upon AC10 depletion both by western blot (WB) and indirect immunofluorescence assay (IFA) (*Figure 2E–F*) providing additional compelling evidence that the two proteins form a complex. Even a short treatment with IAA was sufficient to cause

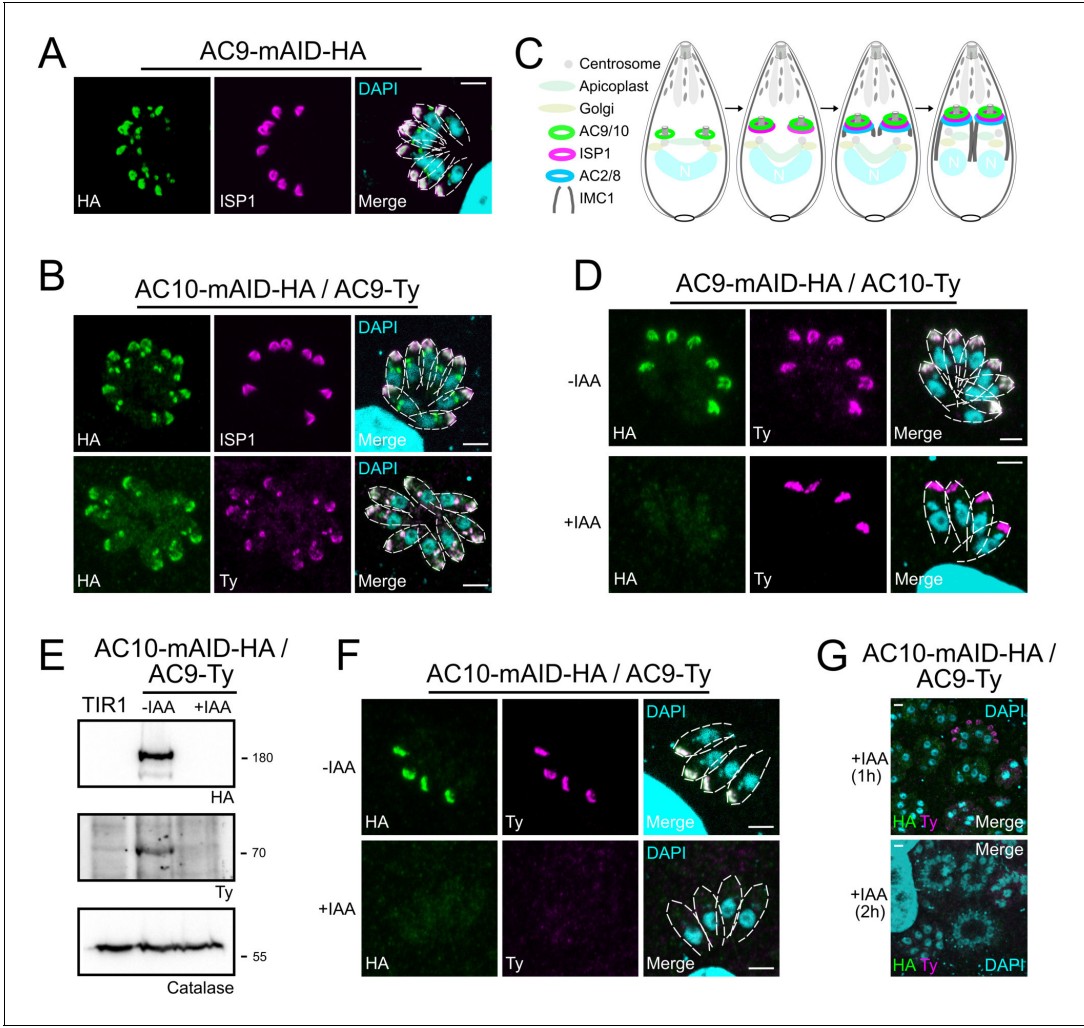

**Figure 2.** AC9 and AC10 are among the earliest markers of parasite division. (**A** and **B**) AC9 and AC10 are both recruited very early during budding of daughter cells and before ISP1. (**C**) Schematic representation of the events during endodyogeny highlighting the sequential insertion of the different components of the IMC/SPN/cytoskeleton. (**D**) AC10 localization was not impacted upon depletion of AC9. (**E** and **F**) Depletion of AC10 caused degradation of AC9. (**G**) Shorter auxin treatments were sufficient to remove AC9 from the apical cap suggesting that AC9 and AC10 form a complex. Scale bars = 2 μm.

The online version of this article includes the following figure supplement(s) for figure 2:

**Figure supplement 1.** No alterations of AC2, AC8 and ISP1 upon depletion of AC9 and AC10.

destabilization of AC9 (*Figure 2G*) without impacting on AC2, AC8 and ISP1 (*Figure 2—figure supplement 1D–F*), yet the apical pole appeared enlarged (arrow) in some parasites (*Figure 2—figure supplement 1C* and *Figure 2—figure supplement 1F*).

## AC9 and AC10 belong to the alveolin network based on super-resolution microscopy

Super-resolution imaging with a stimulated emission depletion (STED) microscope revealed that AC9 and AC10 signals are not homogeneous at the apical cap but rather organized in rows with a regular periodicity (*Figure 3A–B*). Remarkably, IMC1 staining is also organized in regular rows reminiscent of the SPMTs arrangement and the intramembranous particle lattice observed by electron microscopy (EM) (*Morrissette et al., 1997*). Further colocalization of IMC1 with tubulin showed that microtubules are interspaced by two rows of IMC1, a pattern not reported before (*Figure 3—figure supplement 1A*). In contrast, GAP45 staining is more homogeneous along the parasite pellicle on the PM side of the IMC. Colocalization with GAP45 and IMC sub-compartment protein 1 (ISP1) (*Beck et al., 2010*) confirmed that AC9 is confined to the SPMTs side of the IMC and follows a periodic arrangement like SPMTs and alveolins (*Figure 3—figure supplement 1B*). We next applied Ultrastructure Expansion Microscopy (U-ExM) (*Gambarotto et al., 2019*) for the first time in *T. gondii* to gain deeper resolution. As proof of concept, we first used antibodies specific to alpha/beta tubulins and to poly-glutamylation (Poly-E) to detect the SPMTs. The shape and ultrastructure of the parasite were well preserved while the expansion rate was approaching 4x and the SPMTs showed noticeably a high level of poly-glutamylation all along their length, except at their most distal part (*Figure 3—figure supplement 1C*). Interestingly the conoid fibers seem to be devoid of poly-glutamylation. Of relevance, AC9 and AC10 were clearly colocalizing as a regular pattern between each SPMTs just below the conoid and the APR (*Figure 3C–E* and *Figure 3—figure supplement 1D*). Given the close proximity to the SPMTs, AC9 and AC10 were produced recombinantly to assess binding to porcine MTs in an in vitro binding assay (Cytoskeleton ink.). AC9 did not interact directly with these heterologous MTs (*Figure 3—figure supplement 1E*) while recombinant AC10 produced either in bacteria or insect cells formed a gel resistant to solubilization and hence could not be tested. Analysis of the other ACs biotinylated proteins by AC9 revealed that AC2 is localized on the SPMTs up to the APR in contrast to AC8, which is present between the SPMTs and does not reach the APR (*Figure 3F–G*). STED microscopy confirmed that AC9 and AC10 are surrounding the APR stained by RNG2 without being in direct contact with it (*Figure 3H*). Of interest, RNG2 adopted a regular pattern at the APR both by STED and U-ExM, suggesting that the APR is composed of discrete subunits possibly including microtubule plus-end tracking proteins that ensure the apical docking of the SPMTs and their regular interspacing (*Figure 3H–I*).

## AC9 and AC10 are essential for invasion, egress and induced microneme secretion

Upon addition of auxin, AC9 and AC10 are tightly downregulated as shown by IFA (*Figure 4A* and *Figure 4D*) and WB (*Figure 4B* and *Figure 4E*) in the AC9-mAID-HA and AC10-mAID-HA strains. Depletion of both proteins resulted in no lysis plaques in the monolayer of human foreskin fibroblasts (HFF) after 7 days (*Figure 4C* and *Figure 4F*). Further phenotyping showed that depletion of AC9 and AC10 caused a severe defect in invasion (*Figure 4G*) and egress (*Figure 4H*) without impacting on parasite intracellular growth and replication (*Figure 4—figure supplement 1A*). This phenotype is at least in part explained by the severe block in induced microneme secretion observed upon stimulation by two known triggers of microneme exocytosis, BIPPO (a phosphodiesterase inhibitor) (*Howard et al., 2015*) and ethanol (*Carruthers et al., 1999*; *Figure 4I* and *Figure 4—figure supplement 1B*).

## Conditional depletion of AC9 or AC10 causes severe morphological defect of the apical complex

In the absence of AC9 or AC10, numerous proteins crucially implicated in motility, invasion and conoid stability were no longer detectable at the apical complex and notably the conoid-associated motor MyoH (*Graindorge et al., 2016*) and the apical polar ring resident protein RNG2 (*Figure 5A*). Among other apical proteins lost, we found the apical methyltransferase (AKMT) (*Figure 5—figure*

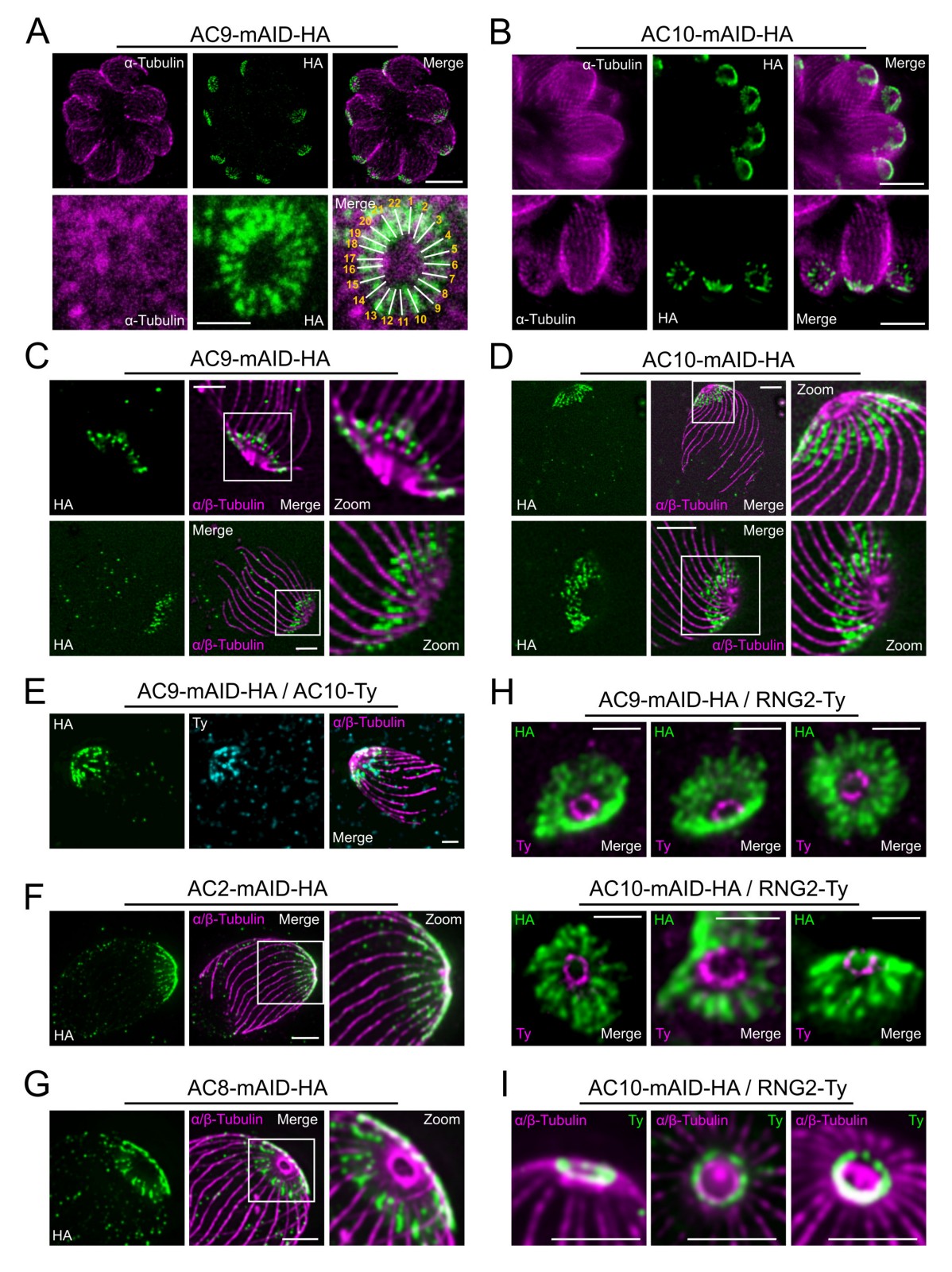

**Figure 3.** Super resolution microscopy revealed a peculiar localization of ACs proteins. (**A** and **B**) Stimulated emission-depletion super-resolution microscopy (STED) revealed that AC9 and AC10 are arranged in periodical rows reminiscent of subpellicular microtubules (SPMTs) arrangement. (**C** and **D**) Ultrastructure Expansion microscopy (U-ExM) confirmed that AC9 and AC10 are arranged in regular rows and localized in between the SPMTs. (**E**) AC9 and AC10 colocalized at the apical cap level. (**F**) AC2 localized on SPMTs by U-ExM at the apical cap up to the APR. (**G**) As AC9 and AC10, AC8

*Figure 3 continued on next page*

Figure 3 continued

localized in between SPMTs. (**H** and **I**) Colocalization with RNG2 revealed that both AC9 and AC10 surrounds the APR without making significant contacts with it. Surprisingly, the RNG2 signal did not appear to be homogeneous at the APR but rather forming discrete dots by STED and U-ExM. Scale bars = 2 μm, except for 3A (bottom panel) and 3H = 0.5 μm.

The online version of this article includes the following figure supplement(s) for figure 3:

**Figure supplement 1.** STED and U-ExM revealed unprecedented localization of alveolin network proteins and tubulin modification.

supplement 1A; *Heaslip et al., 2011*), the conoid ankyrin-repeat containing protein hub 1 (CPH1) (*Figure 5—figure supplement 1B*) and the apical actin nucleator FRM1 (*Figure 5—figure supplement 1C*; *Tosetti et al., 2019*). Conversely, ICMAP1, a MTs binding protein localizing to the intraconoidal microtubules (*Heaslip et al., 2009*) was still detectable but mispositioned (*Figure 5—figure supplement 1D*). Remarkably, all the proteins lost in the conoid of the mature parasite (mother) were still present in the forming daughter cells suggesting that these markers were lost during the last step of daughter cell formation.

Deoxycholate, a detergent which solubilizes the membranes (PM and IMC), was used shortly on extracellular parasites to visualize the cytoskeleton of the parasites by EM. Strikingly, AC9 treated parasites resulted in the loss of the APR and conoid, thus explaining the disappearance of apical markers observed by IFA (*Figure 5B*). Focus Ion Beam Scanning Electron Microscopy (FIB-SEM) on intracellular parasites compellingly revealed that the conoid is absent in mature parasites while it is still visible in daughter cells, thus explaining the loss of apical markers only after completion of parasite division (*Figure 5C* and *Video 1*). The accumulation of micronemes at the very tip of the parasites was absent in AC9 depleted parasites and in some cases gave the impression that the organelles were in the vacuolar space (*Figure 5—figure supplement 1E*). Consistent with the sporadic vacuolar staining observed with anti-MIC2 antibodies, some parasites displayed an apical dilatation of the PM into the PV space by EM (*Figure 5—figure supplement 1F*). Moreover, rhoptries were still attached to parasite apex but less bundled compared to untreated parasites (*Figure 5—figure supplement 1G*).

## The conoid and conoid-associated proteins are lost during division in the absence of AC9

To confirm that the loss of the conoid occurred specifically during division, time point experiments were conducted. Importantly, 2 hr of incubation in presence of IAA were sufficient to degrade AC9 by WB in intracellular parasites (*Figure 6—figure supplement 1A*) and by IFA on extracellular parasites (*Figure 6—figure supplement 1B*). Non-dividing extracellular parasites treated overnight with IAA showed no loss of apical markers as illustrated by staining of MyoH (*Figure 6A*). In contrast, intracellular parasites treated with IAA at different time points, showed that MyoH disappeared from the conoid only in parasites that had completed at least one cycle of division, that is after at least 4 hr of IAA treatment (*Figure 6B*). Concordantly, extracellular parasites treated with IAA were not affected in microneme secretion (*Figure 6C*) whereas treatment on intracellular parasites led to a gradual increase in microneme secretion defects over time (*Figure 6—figure supplement 1C*). The conoid markers analyzed so far are incorporated in the daughter cells since the beginning of the division process, shortly after centrosome duplication. Next, we assessed the fate of APR markers such as kinesinA (KinA) and APR1, which are incorporated very early (*Leung et al., 2017*) and RNG1 a late marker of division (*Tran et al., 2010*). In the absence of AC9 or AC10, APR1 and KinA are lost in mature parasites, but present in daughter cells, respectively (*Figure 6D–E*). We confirmed the late incorporation of RNG1 in daughter cells (*Figure 6F–G*). In presence of IAA, RNG1 failed to localize at the APR when daughter cells were emerging from the mother cell, resulting in disorganized dots (*Figure 6F–G* and *Figure 6—figure supplement 1D–E*). Those observations confirmed that the disappearance of the APR, like the conoid, occurs at the final stage of the division process. Interestingly, IFA performed on intracellular parasites fixed earlier post-invasion revealed that the conoid is not lost in a synchronous manner, at least during the first 2 division cycles (*Figure 6—figure supplement 1F*). Some apical markers could still be observed at the very late stage giving some hints about a possible explanation of the loss of conoid and APR occurring at the end of division and/or

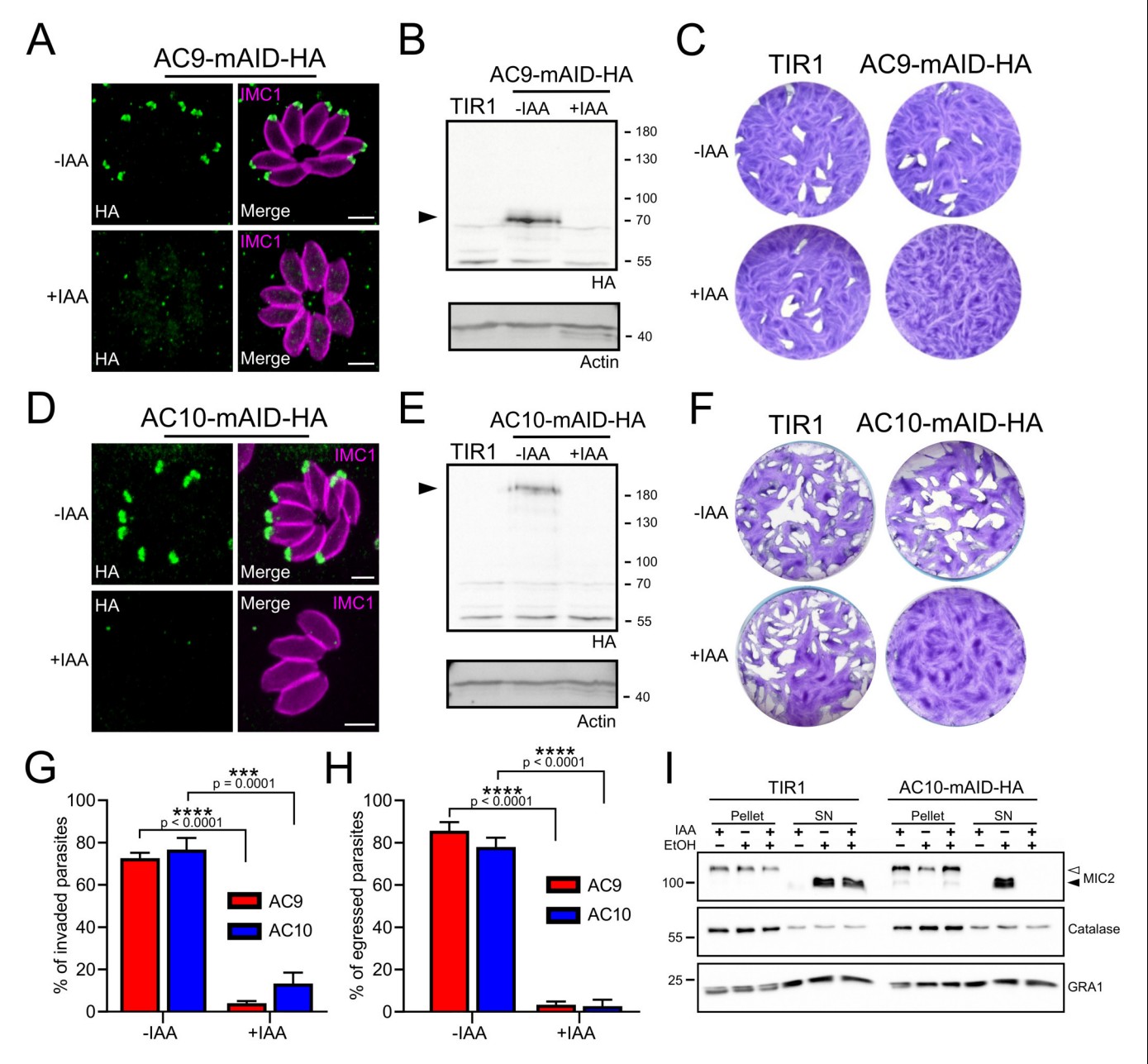

**Figure 4.** AC9 and AC10 are essential proteins for microneme secretion, invasion and egress. (A), (B) and (C) AC9-mAID-HA was tightly regulated by addition of IAA and AC9 depleted parasites failed to form lysis plaques after 7 days. TIR1 represents the parental strain and was used as a control. (D), (E) and (F) AC10-mAID-HA was rapidly degraded upon addition of IAA and its depletion resulted in no visible plaque formation after 7 days. (G and H) Both invasion and egress were severely impaired in absence of AC9 and AC10. Data represented are mean values ± standard deviation (SD) from three independent biological experiments. A parametric paired t-test was used to assess significance; the two-tailed p-values are written on the graphs. (I) Microneme secretion was completely abolished in AC10 depleted parasites stimulated by ethanol. Pellets and supernatants (SN) were analyzed using α-MIC2 antibodies for secretion (white arrow: full length MIC2; black arrow: secreted MIC2), α-catalase (CAT) to assess parasites lysis and α-dense granule 1 (GRA1) for constitutive secretion. Scale bars = 2 μm.

The online version of this article includes the following figure supplement(s) for figure 4:

**Figure supplement 1.** AC9 and AC10 depletion did not cause any defect in parasite replication.

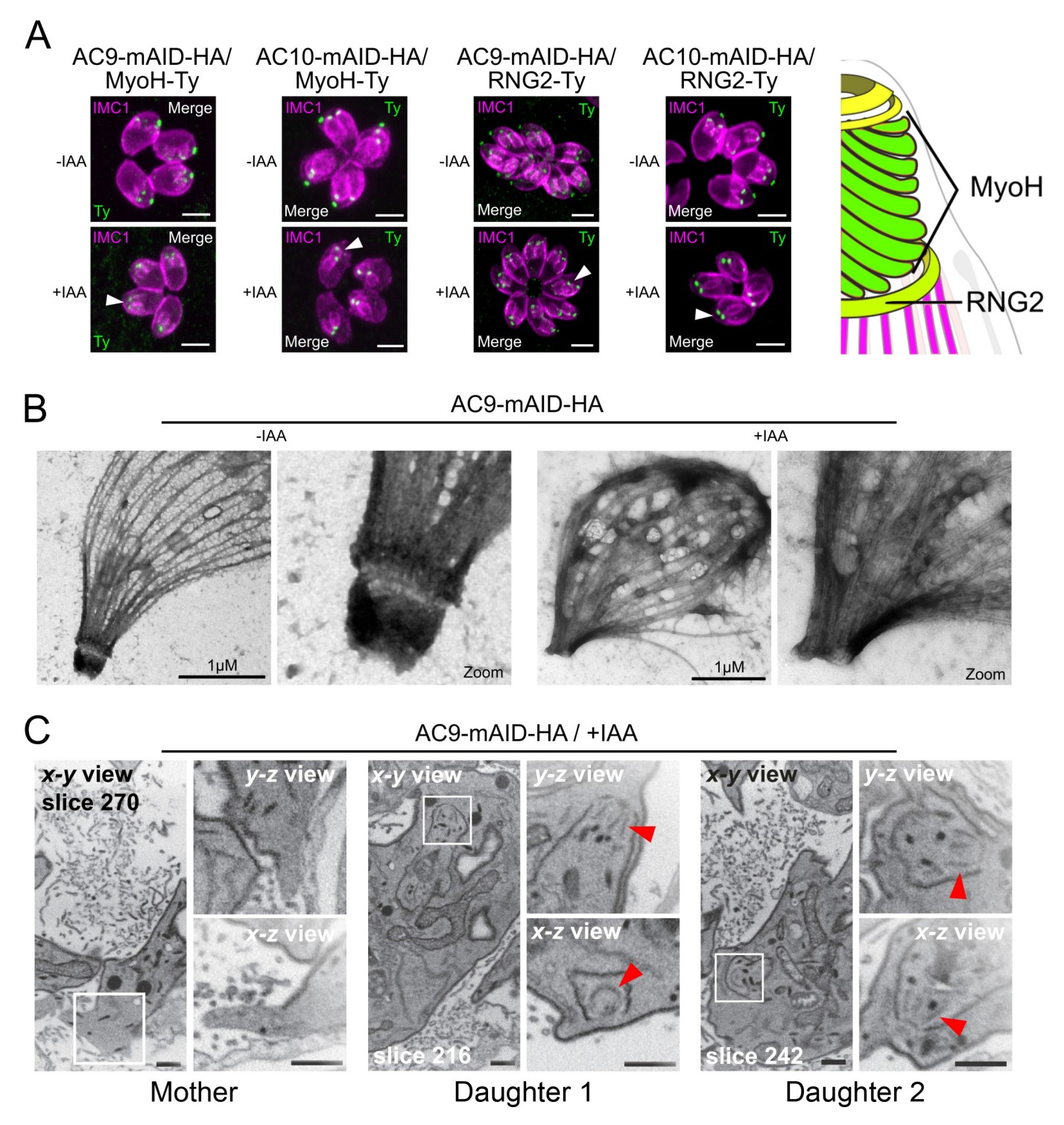

**Figure 5.** AC9 and AC10 depleted mother parasites lost the APR and the conoid. (**A**) Both conoid-associated MyoH and apical polar ring RNG2 staining were completely lost in mature parasites while still present in forming daughter cell (arrowhead) in absence of AC9 and AC10. (**B**) AC9 depletion caused loss of the conoid and the apical polar ring in agreement with IFA analysis using RNG2 marker. Parasite were grown for 48 hr -/+ IAA, syringe lysed and processed as described in Materials and methods. (**C**) FIB-SEM confirmed that loss of AC9 impacted the conoid and APR exclusively in the mother cell. Red arrows highlight conoids. Scale bars = 2 μm.

The online version of this article includes the following figure supplement(s) for figure 5:

**Figure supplement 1.** Destabilization of apical complex components upon depletion of AC9.

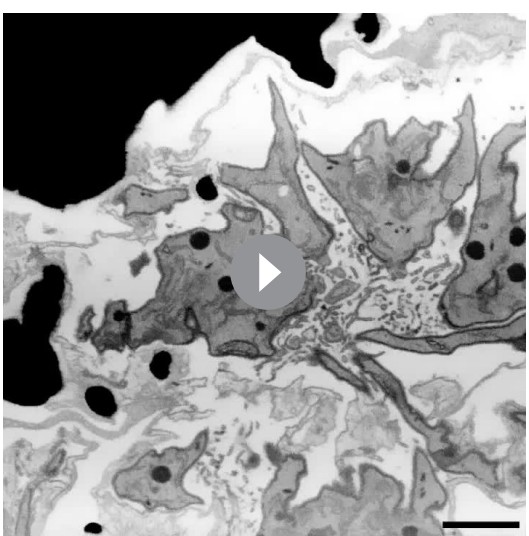

**Video 1.** FIB-SEM analysis highlighted the specific loss of conoid in the mother cell but not in the growing daughter cells.

https://elifesciences.org/articles/56635#video1

daughter cells emergence, when newly formed parasites acquire the PM from the mother (*Figure 6—figure supplement 1F*).

## AC9 and AC10 are required for the organization of the SPMTs at the apical cap

Given the periodicity of AC9 and AC10 arrangement between the SPMTs and the loss of both conoid and APR upon depletion of two proteins, we wondered if the overall microtubular structure was also impacted. Deoxycholate extraction revealed that in the AC9 and AC10 depleted parasites, the SPMTs were disconnected (*Figure 7A–B*) and only in rare cases the overall structure was maintained despite the obvious absence of APR and conoid (*Figure 7—figure supplement 1A*). When applying U-ExM in AC9 and AC10 depleted parasites, loss of APR and conoid were confirmed and the apex appeared enlarged. Remarkably, SPMTs appeared significantly disorganized at the apical pole of mature parasites (*Figure 7C–F* and *Figure 7—figure supplement 1B*). Of relevance, the depletion of AC9 resulted in the loss of the preconoidal rings while the peripheral annuli were not affected, when visualized by Centrin2 labeling (*Figure 7F*).

## Discussion

In this study we have identified two apical cap proteins implicated in the stability of the APR and the anchoring of the conoid to the apical complex of mature parasites. AC10 was identified as the most prominent partner of AC9 via proximity biotinylation of AC9-BirA. Conditional depletion of AC9 or AC10 resulted in a very severe phenotype with parasites impaired in microneme secretion and consequently unable to glide, invade and egress from infected cells. Mature parasites lacking AC9 and AC10 are conoidless and also deprived of APR but replicate normally. Moreover, in the absence of conoid, FRM1, MyoH, AKMT and GAC are missing. In consequence no actin filament is produced and delivered to the glideosome, explaining the extreme severity of the phenotype.

### Several ACs are components of the alveolin network conserved in coccidians

Super-resolution microscopy by STED and U-ExM unambiguously established that AC9 and AC10 are not distributed evenly in the apical cap region but organized in longitudinal rows intercalating between the 22 spiraling SPMTs. Fractionation assays indicated that both proteins are poorly soluble, consistent with their implication in the alveolin network that spreads between the IMC and the SPMTs. The lack of solubility of AC10 hampered biochemical demonstration that AC9 and AC10 form a complex. However, their perfect colocalization and the fact that AC10 depletion phenocopied AC9 and caused its destabilization strongly support their functional and physical association. In addition to AC9 and AC10, AC5 (or TLAP3) was previously suggested to arrange in such longitudinal rows in the apical cap region (*Liu et al., 2016*). AC9 and AC10 are among the earliest markers of parasite division, prior to ISP1, AC1 and AC5. Like AC9 and AC10, AC8 localizes between SPMTs without being in contact with the APR. In contrast AC2 is coating the SPMTs up to the APR suggesting that it is a microtubule-associated protein (MAPs). It would be informative to assess the precise localization of the other ACs even if the corresponding genes are not predicted to be essential (*Chen et al., 2015*) to comprehensively define the composition of the alveolin network. Of interest, the first described alveolin protein IMC1 is also arranged in longitudinal rows and not homogeneously distributed around the parasite (*Mann and Beckers, 2001*). IMC1 appears to form two

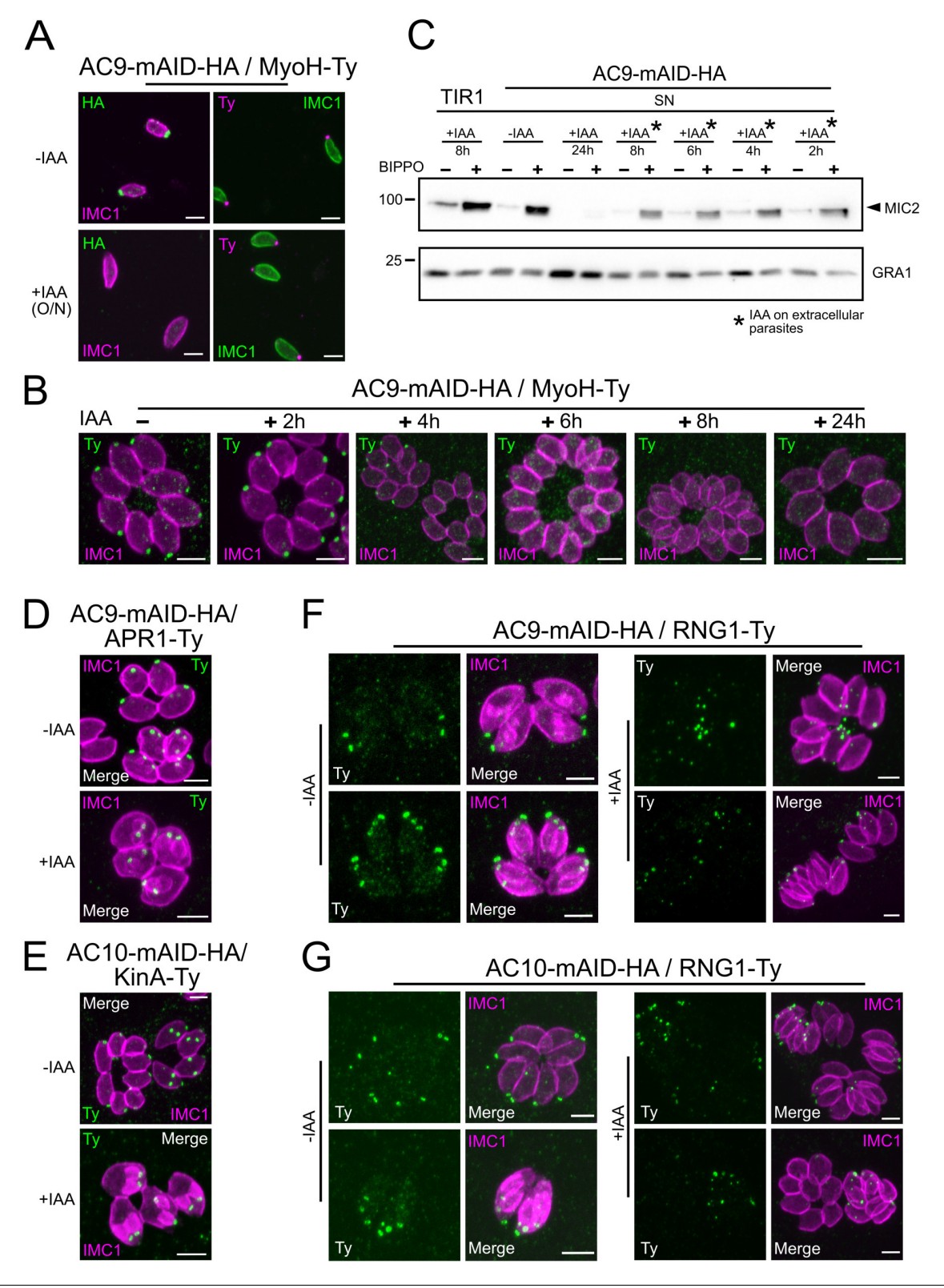

**Figure 6.** Loss of the APR and conoid occur during the last stages of division. (**A**) IAA overnight treatment of extracellular parasites did not result in MyoH loss from the apical tip of non-dividing mature cell. (**B**) MyoH is lost only during parasite division after more than 2 hr treatment (i.e. the time to complete at least one division cycle). (**C**) IAA was added at different time point on extracellular parasites showing that microneme secretion is not affected in such conditions. (**D** and **E**) APR1 and KinA are lost in mature parasites upon deletion of AC9 and AC10, respectively. (**F** and **G**) RNG1 is

*Figure 6 continued on next page*

Figure 6 continued

incorporated very late in daughter cell, way after the alveolin network scaffold stained with IMC1. Upon depletion of both AC9 and AC10, RNG1 failed to be properly anchored at the APR in most parasites. Scale bars = 2 μm.

The online version of this article includes the following figure supplement(s) for figure 6:

**Figure supplement 1.** Loss of the APR and conoid just before the emergence of daughter cells.

rows between each SPMT, again reminiscent of the double rows of IMPs observed by freeze-fracture studies (*Morrissette et al., 1997*). IMC1 is well conserved (*Dubey et al., 2017*) and harbors a strong negative fitness score, however its function in *T. gondii* has not been reported. Of note, AC9 and AC10 are conserved uniquely in the cyst forming coccidian subgroup of Apicomplexa that includes *T. gondii*, *Neospora caninum*, *Besnoitia besnoiti* and *Cystoisospora suis.* Both proteins are absent in *Eimeria* spp., piroplasmida, haemosporidia and cryptosporidia, while other ACs appear to be conserved in other lineages including distantly related gregarines and chromerids. To determine if AC9 and AC10 could potentially interact with other alveolar proteins or MAPs to stabilize the apical complex we examined the localization of ISP1, AC2, AC3 and AC8 in parasites depleted in AC9 or AC10 and saw no alteration.

## Conoid and APR are lost as consequence of SPMTs disorganization in the absence of AC9 or AC10

The cascade of events leading to the loss of the conoid and APR involves parasite division. Strikingly, the conoid and APR are present in growing daughter cells and are only lost at the final stage of endodyogeny, when the daughter cells emerge and their cytoskeleton confers shape and rigidity to the parasite. The detached conoid appears to be very rapidly degraded because we fail to detect it inside mature parasites even when scrutinizing with FIB-SEM. However, with IFA taken at early stage on parasites that have completed only one or two division cycles, some degrading conoids could be observed with MyoH used as marker. In larger vacuoles such staining were not observed because the turnover and recycling of materials in the enlarged residual body might be faster. Contrastingly, the daughter cells were always shown to possess the conoid. In deoxycholate extracted AC9 and AC10 depleted parasites, the SPMTs are formed properly but are not joint together by the APR, which is absent in these mutants. A similar but milder phenotype was also reported in the double knock-out of KinA and APR1 with SPMTs partially detached from the parasite apex because of the fragmentation of the APR (*Leung et al., 2017*). The strategic position of AC9 and AC10 intercalated between SPTMs suggested a possible implication in maintaining the correct interspacing between the SPMTs. Of note, the stability of SPMTs is in part maintained by the glideosome-associated protein with multiple-membrane spans family 1 (GAPM1) and other members of the GAPM family (*Harding et al., 2019*).

Here we have exploited the U-ExM technology to examine the SPMTs at an unprecedented level of resolution using anti-tubulin as well as anti-polyglutamylated (PolyE) antibodies. Of relevance the SPTMs are more glutamylated toward the apical pole of the parasites suggesting that the modification accumulates over time and is absent in the atypical tubulin fibers that form the conoid. The application of U-ExM on the AC9 and AC10 depleted mutants revealed a very striking apical disorganization of the SPMTs. The resulting disordered microtubules could cause the destabilization of the APR, which is present at the top of the SPMTs, presumably serving as MTOC. The dramatic consequences of AC9 and AC10 depletion on SPTMs, APR and conoid occur very late during parasite division (*Figure 8*). Concordantly, no major perturbations on the overall shape of the parasite, the IMC, the annuli and the apical positioning of microneme and rhoptries were observed. At the time when the daughter cells emerge from the mother cell, the tension imposed on the SPMTs could plausibly lead to their disorganization and consequently to the loss of APR and conoid and hence explains why the phenotypes of AC9 and AC10 depletion only appear at the end of the division process.

Taken together, these findings lead us to postulate that the loss of the conoid is presumably a consequence of the loss of the APR, which could result from the disorganization of the SPMTs at the apical pole of the parasite. In such a model, the AC9/AC10 complex would act as a glue in between SPMTs and establish a functional link between the alveolin network and the SPMTs.

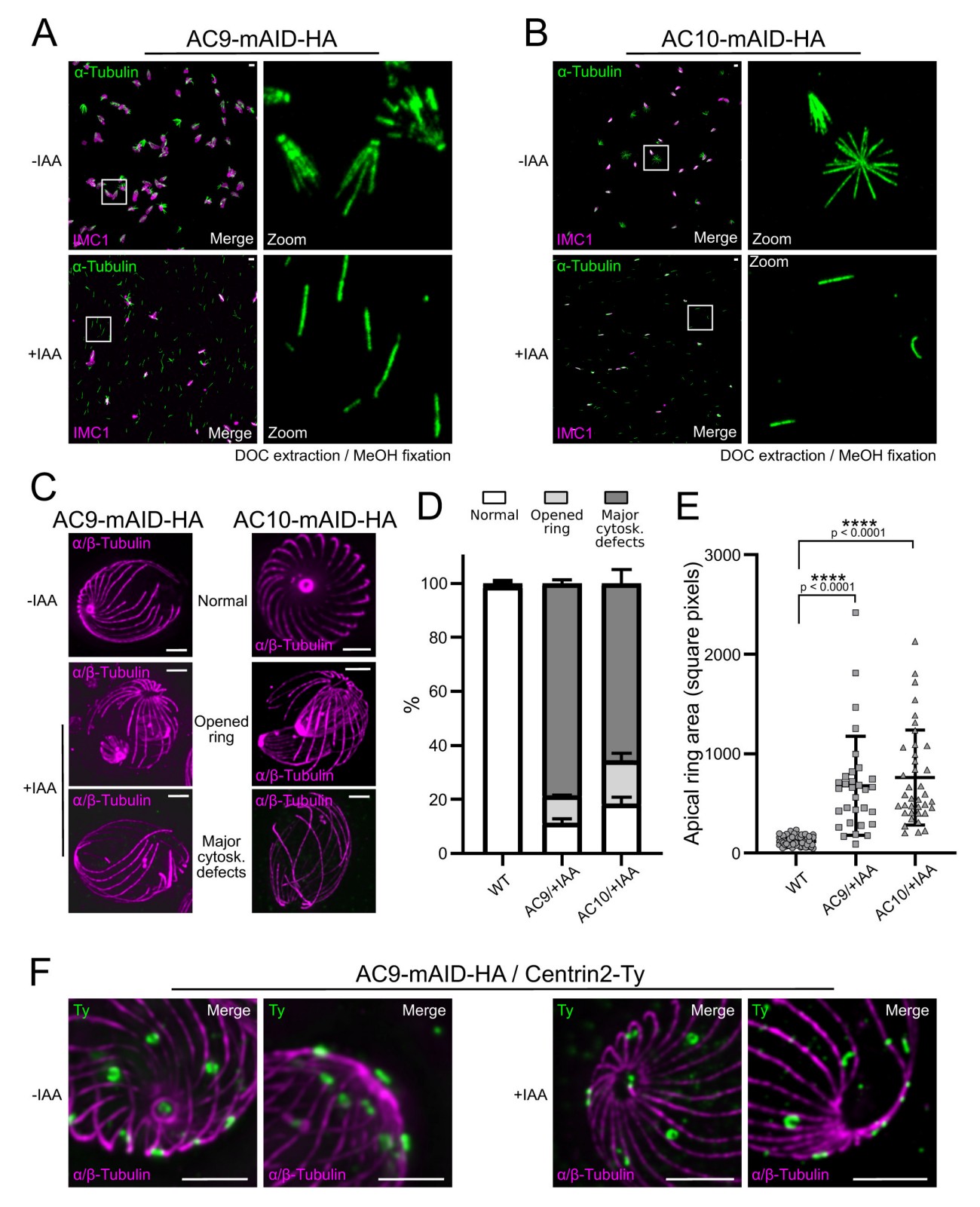

**Figure 7.** Depletion of AC9 and AC10 caused major cytoskeletal defects. (**A** and **B**) Freshly egressed parasites were extracted with DOC on gelatin coverslips. In AC9 and AC10 depleted parasites the microtubular cytoskeleton collapsed and only single microtubules can be visualized. (**C**) U-ExM confirmed that parasites depleted of AC9 and AC10 presented an enlarged apical opening while the microtubular cytoskeleton presented major structural defects. (**D**) Quantification of the phenotypical defects displayed in panel (**C**). Two independent preparations show statistically significant

*Figure 7 continued on next page*

*Figure 7 continued*

differences between the WT group and the AC9/+IAA and AC10/+IAA groups for each of the three categories (normal, open ring, major cytoskeletal defects). Chi2 tests, standardized deviation from expected values and Fisher's test were performed to ensure the statistical differences between each categories of the WT group versus AC9/+IAA or AC10/+IAA groups. (*Figure 7—source data 1*) (E) Measure of the area of the apical polar ring show statistically significant differences between the three groups. A non-parametric test (ANOVA) was performed to ensure statistical significance. Tukey all pairs comparison test was used to assess significance between groups (p-values are shown on the figure) (*Figure 7—source data 2*) (F) AC9 depletion caused the loss of pre-conoidal rings while the peripheral annuli were not compromised. Scale bars = 2 μm.

The online version of this article includes the following source data and figure supplement(s) for figure 7:

**Source data 1.** Statistical analysis for the *Figure 7D*.
**Source data 2.** Statistical analysis for the *Figure 7E*.
**Figure supplement 1.** Structural defects upon depletion of AC9 and AC10.

## Materials and methods

### Accession numbers

AC9 (TGGT1_246950) and AC10 (TGGT1_292950). All resources are listed in Key Resources Table (*Supplementary file 3*).

### Parasite culture

T. gondii tachyzoites were grown in human foreskin fibroblasts (HFFs, American Type Culture Collection-CRL 1634) in Dulbecco's Modified Eagle's Medium (DMEM, Gibco) supplemented with 5% fetal

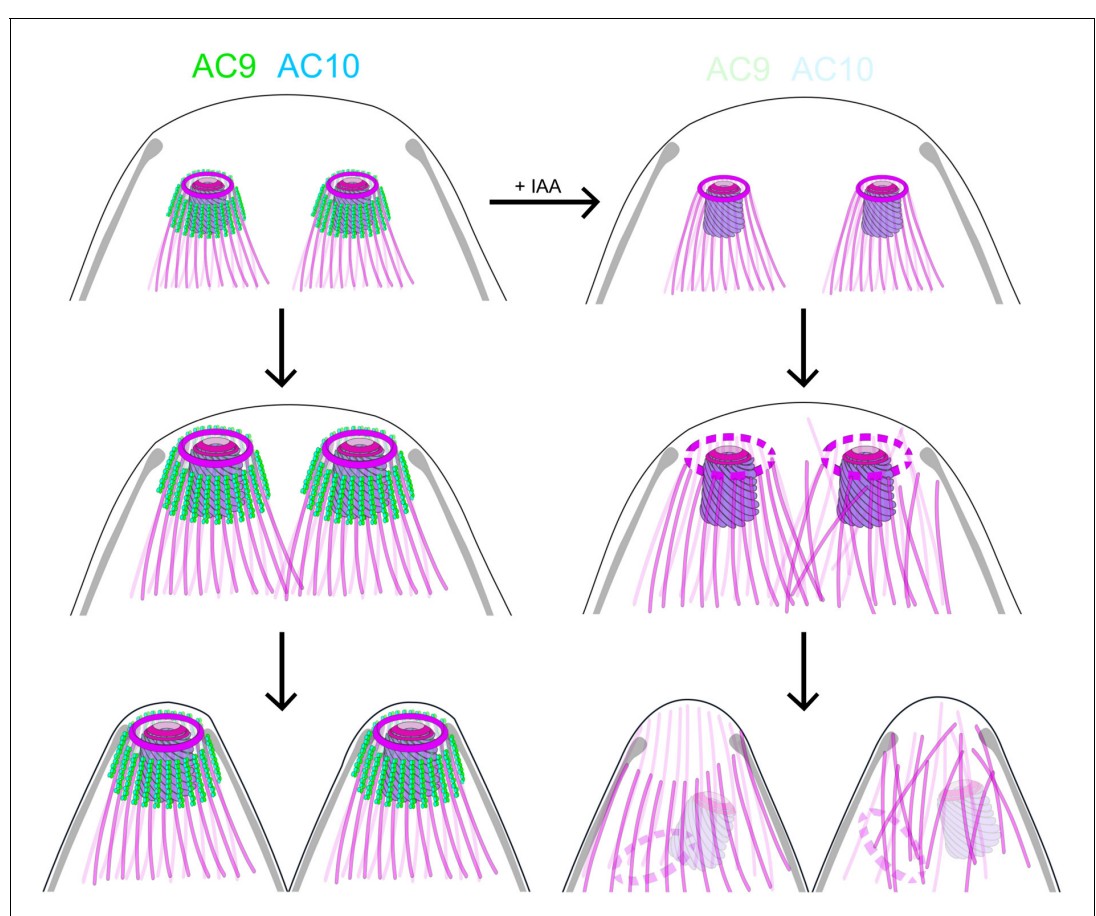

**Figure 8.** Role of AC9 and AC10 in the stability of the apical complex. Schematic representation of AC9 and AC10 and their role in maintaining the integrity of the structural components of the apical complex during the late stage of parasite division.

calf serum (FCS), 2 mM glutamine and 25 μg/ml gentamicin. Depletion of AC9-mAID-HA and AC10-mAID-HA were achieved with 500 μM of auxin (IAA) (*Brown et al., 2017*).

## Cell lines

Human Foreskin Fibroblast (HFF) cells were purchased from the ATCC (CRL-2088 Ref# CCD1072Sk). ATCC guarantees their origin and ensure that they are mycoplasma free. Cultured cells were screened regularly against mycoplasma contamination by PCR and immunofluorescence assay.

## Cloning of DNA constructs

Genomic DNA extractions were performed with the Wizard SV genomic DNA purification kit (Promega). PCRs were performed with Q5 (New England Biolabs) and KOD (Novagen) polymerases. Primers used are listed in the *Supplementary file 4*. Cloning were performed with *E. coli* XL-1 Gold chemo-competent bacteria. For endogenous epitope tagging, genomic DNA of the C-terminus of CPH1, RNG2 and ICMAP1 were amplified by PCR, digested with restriction enzymes and cloned into ASP5-3Ty-DHFR (*Supplementary file 4*; *Hammoudi et al., 2015*). Specific gRNA and KOD PCR were generated using the Q5 site-directed mutagenesis kit (New England Biolabs) on the pSAG1:: Cas9-U6::sgUPRT vector (*Shen et al., 2014*) for all the other KI constructs listed in *Supplementary file 4*. AC9-mAID-HA and AC10-mAID-HA were generated by KOD PCR as described in *Brown et al., 2018*; primers for KOD and gRNA are listed in *Supplementary file 4*.

## Parasite transfection and selection of transgenic parasites

*T. gondii* tachyzoites were transfected by electroporation as previously described (*Soldati and Boothroyd, 1993*). Mycophenolic acid (25 mg/mL) and xanthine (50 mg/mL) or pyrimethamine (1 μg/ml) were employed to select resistant parasites carrying the HXGPRT and the DHFR cassette, respectively.

## Antibodies

The antibodies employed are the following: rabbit polyclonal: α-GAP45 (*Plattner et al., 2008*), α-IMC1 (*Frénal et al., 2014*), α-ARO (*Mueller et al., 2013*), α-GAC (*Tosetti et al., 2019*), α -polyglutamate chain (PolyE, IN105) (1:500; AG-25B-0030-C050, AdipoGen) and α-HA (Sigma). Mouse monoclonal: α-ACT (*Herm-Götz et al., 2002*), α-ISP1 (8), α-SAG1, α-Ty, α-MIC2, ROP2-4 (gifts from J-F Dubremetz, Montpellier) and acetylated α-tubulin (6-11B-1; Santa Cruz Biotechnology). Tubulin antibodies AA344 scFv-S11B (β-tubulin) and AA345 scFv-F2C (α-tubulin). Secondary antibodies Alexa Fluor

405-, Alexa Fluor 488-, Alexa Fluor 594-conjugated goat α-mouse/α-rabbit, were used for IFA. For western blot revelation, secondary peroxidase conjugated goat α-rabbit or mouse antibodies (Sigma) were used.

## Immunofluorescence assay

Parasites were inoculated on HFF cells with cover slips in 24-well plates, grown for 24–30 hr, fixed with either 4% PFA/0.05% glutaraldehyde (PFA/GA) or cold methanol, neutralized in 0.1M glycine/ PBS for 5 min and processed as previously described (*Plattner et al., 2008*).

## Confocal microscopy and super-resolution microscopy (STED)

Confocal images were obtained with a Zeiss laser scanning confocal microscope (LSM700 using objective apochromat 63x/1.4 oil) and Leica TCS SP8 STED 3X. Experiments were conducted at the Bioimaging core facility of the Faculty of Medicine at University of Geneva. Confocal images were then processed with ImageJ while STED pictures with LAS X.

## Electron microscopy

For negative staining, extracellular parasite either not treated with IAA or treated with IAA were pelleted in PBS. Conoid protrusion was induced by incubation with 40 μL of BIPPO in PBS for 8 min at 37°C. 4 μL of the sample were applied on glow-discharged 200-mesh Cu electron microscopy grid for 10 min. The excess of the sample was removed by blotting with filter paper and immediately washed on drop of double distilled water. Then, the parasite cytoskeleton was extracted by

incubation on two 50 μL droplets of 10 mM deoxycholate in humidified chamber protected from light for 2 × 5 min. Excess of the detergent was removed by 3 washes on drops of double distilled water and finally the sample was negatively stained with 1% aqueous solution of uranyl acetate for 20 s and air dried. Electron micrographs of protruded conoid were collected with Tecnai 20 TEM (FEI, Netherland) operated at 80 kV acceleration voltage equipped with side mounted CCD camera (MegaView III, Olympus Imaging Systems) controlled by iTEM software (Olympus Imaging Systems). FIB-SEM samples were prepared as extensively described in *Hammoudi et al., 2018*.

## Invasion assay
Freshly egressed parasites were inoculated on HFF cells with cover slips in 24-well plates, centrifuged for 1 min at 1000 rpm and placed at 37°C for 30 min before fixing for 7 min with PFA/GA. Fixed cells were incubated first in 2% BSA/PBS for 30 min, then with α-SAG1 antibodies for 20 min and washed 3 times with PBS. Next cells were fixed with 1% formaldehyde for 7 min, washed once with PBS and permeabilized with 0.2% Triton X-100/PBS for 20 min. Cells were incubated with α-GAP45 antibodies, washed 3 times and incubated with secondary antibodies. 200 parasites were counted for each condition. Data represented as mean values ± standard deviation (SD) (three independent biological experiments).

## Egress assay
*T. gondii* tachyzoites were grown for 30 hr on HFF cells with cover slips in 24-well plates. The infected host cells were incubated for 7 min at 37°C with DMEM containing BIPPO prior to fixation with PFA/GA. Immunofluorescence assays were performed as previously described with α-GAP45 antibodies and 200 vacuoles were counted. Data are presented as mean values ± SD (three independent experiments).

## Plaque assay
HFFs were infected with fresh parasites and grown for 7 days before fixation with PFA/GA. After fixation, HFFs were washed with PBS and the host cells monolayer was stained with crystal violet.

## Intracellular growth assay
Parasites were grown for 30 hr in 24-well plates prior to fixation with PFA/GA for 10 min. IFA with α-GAP45 antibodies was performed as previously described and the number of parasites per vacuole was counted (200 vacuoles). Data are mean values ± SD (three independent biological experiments).

## Microneme secretion
Freshly egressed parasites were harvested after ~48 hr of grown and pellets washed twice in 37°C pre-warmed intracellular buffer (5 mM NaCl, 142 mM KCl, 1 mM $MgCl_2$, 2 mM EGTA, 5.6 mM glucose and 25 mM HEPES, pH 7.2). Parasites were incubated at 37°C for 15 min in DMEM containing either Ca2+ ionophore, BIPPO or propranolol. Parasite were centrifuged at 1000 g for 5 min at 4°C and supernatants (SN) transferred in different tubes. Pellets were washed once in PBS while supernatants were centrifuged again at higher speed (2000 g) for 5 min at 4°C to clean samples from parasite debris. Pellets and SNs were analyzed by Western blot using α-MIC2, α-catalase (CAT) and α-dense granule 1 (GRA1) antibodies.

## Deoxycholate extraction
Freshly egressed parasites were attached to Poly-L-Lysine-coated coverslips and treated with 10 mM deoxycholate for 20 min at room temperature. Parasites were fixed with methanol for 8 min and proceeded as for IFAs as previously described. Intracellular parasites were treated with IAA for 24 hr prior to experiment.

## Fractionation assay
Freshly egressed tachyzoites were harvested, washed in PBS and then resuspended in either PBS, PBS/1% Triton-X-100, PBS/0.1 M Na2CO3, pH 11.5 or SDS. Parasites were lysed by 3 cycle of freeze and thaw in liquid nitrogen and incubated on ice for 30 min. Pellets and supernatants were separated by centrifugation at 4°C for 30 min at 15000 rpm.

## Expression and purification of recombinant AC9 protein from bacteria

Full-length AC9 protein, containing a N-terminal His-10 tag, was subcloned into bacterial expression vector. One and a half liter of BL21 cells expressing His10-AC9 were grown until reaching an OD600 of 0.6. Protein expression was induced by adding 0.5 mM IPTG and leaving the cells to grow for 70 hr at 12°C. Cells were harvested and the pellet was resuspended in 50 ml of lysis buffer (50 mM Tris pH 8, 500 mM NaCl, 15 mM imidazole, 1 mM Triton X-100, 5% glycerol, 5 mM beta-mercaptoethanol). Cells were lysed by two passages through a French Press at 1000 bar, followed by a brief sonication. Cell debris and membranes were removed by centrifugation at 40'000 g for 35 min at 4°C. Soluble fraction was passed through a 5 ml His Trap column equilibrated in lysis buffer. After loading, the column was washed with 50 ml of lysis buffer, followed by 30 ml of buffer A (20 mM Tris pH 8, 400 mM NaCl, 15 mM imidazole, 3 mM beta-mercaptoethanol). Protein was eluted with buffer A supplemented with 400 mM imidazole. The eluted protein was concentrated to 1 ml using AMICON 30 MWCO concentrator and applied to a Superdex 200 10/300 increase GL size-exclusion chromatography column equilibrated in buffer C (20 mM Tris pH 7.4, 150 mM NaCl, 3 mM DTT). The pure protein eluted after a volume of 10 ml. Fractions containing pure AC9 were pooled and concentrated to 5 mg/ml before flash freezing in liquid nitrogen.

## Microtubules binding assay

The microtubules binding experiments were performed with the Microtubule Binding Protein Spin-Down Assay Biochem Kit (Cytoskeleton, Inc) following manufacturer protocol.

## U-ExM

Parasites were centrifuged at 1000 rpm during 5 min at 32°C and resuspended in 300 μL of PBS 1X. Parasites were sedimented on poly-lysine (A-003-E, SIGMA) coverslips (150 μL/coverslip) during 10 min at RT. Then parasites were fixed in −20°C methanol during 7 min and prepared for Ultrastructural Expansion Microscopy (U-ExM) as previously published (*Gambarotto et al., 2019*). Briefly, coverslips were incubated for 5 hr in 2 × 0.7% AA/1% FA mix at 37°C prior gelation in APS/Temed/Monomer solution (19% Sodium Acrylate; 10% AA; 0,1% BIS-AA in PBS 10X) during 1 hr at 37°C. Then denaturation was performed during 1h30 at 95°C (*Le Guennec et al., 2020*). Gels were expanded overnight in water and after shrinking in PBS gels were stained 3 hr at 37°C with primary antibodies against HA-tag (1:200), Ty-Tag (1:2.5), PolyE (1:500) and α-tubulin and β-tubulin (1:200). Gel were washed 3 × 10 min in PBS-Tween 0.1% prior incubation with secondary antibodies (anti-mouse Alexa 488, anti-mouse Alexa 568, anti-mouse STAR RED anti-rabbit Alexa 488, Anti-mouse Alexa 568, Anti-guinea pig Alexa 568, Anti-guinea pig Alexa 594) (1:400) during 3 hr at 37°C and 3 washes of 10 min in PBS-Tween. Overnight a second round of expansion was done in water before imaging. Imaging was performed on a Leica Thunder inverted microscope using 63 × 1.4 NA oil objective with Small Volume Computational Clearing mode to obtained deconvolved images. 3D stacks were acquired with 0.21 μm z-interval and x,y pixel size of 105 nm. Images were analyzed and merged using ImageJ software.

## Acknowledgements

We thank the Bioimaging Core Facility, Proteomic Core Facility and Proteins/Peptides Platform (University of Geneva – CMU) for their excellent technical support. We thank the MeBoP students of 2019 for their help in replicating and discussing some of the results presented in this work. We thank Oksana Fiammingo for her excellent technical support, Oscar Vadas for protein expression and purification at the platform of the CMU and Marine Laporte for her critical help with the statistical analysis. We thank Damien Jacot for its critical reading of the manuscript. This work was supported by the European Research Council (ERC) under the European Union's Horizon 2020 research and innovation program under grant agreement no. 695596 to DSF, the Swiss National Science Foundation (SNSF) 310030_185325 to DSF and PP00P3_187198 to PG, Novartis Foundation for medical-biological Research and ERC ACCENT StG 715289 to PG and the EMBO long term fellowship 284–2019 to EB.

## Additional information

### Competing interests

Dominique Soldati-Favre: Senior editor, *eLife*. The other authors declare that no competing interests exist.

### Funding

| Funder | Grant reference number | Author |
|---|---|---|
| Schweizerischer Nationalfonds zur Förderung der Wissenschaftlichen Forschung | 310030_185325 | Dominique Soldati-Favre |
| H2020 European Research Council | 695596 | Dominique Soldati-Favre |
| Schweizerischer Nationalfonds zur Förderung der Wissenschaftlichen Forschung | PP00P3_187198 | Paul Guichard |
| Novartis Foundation for Medical-Biological Research | | Paul Guichard |
| European Research Council | ACCENT StG 715289 | Paul Guichard |
| EMBO | 284–2019 | Eloïse Bertiaux |

The funders had no role in study design, data collection and interpretation, or the decision to submit the work for publication.

### Author contributions

Nicolò Tosetti, Conceptualization, Data curation, Formal analysis, Investigation, Methodology, Writing - original draft; Nicolas Dos Santos Pacheco, Conceptualization, Data curation, Formal analysis, Investigation, Methodology, Writing - review and editing; Eloïse Bertiaux, Investigation, Visualization, Methodology; Bohumil Maco, Formal analysis, Investigation, Visualization, Methodology; Lorène Bournonville, Investigation, Methodology; Virginie Hamel, Supervision, Methodology; Paul Guichard, Data curation, Formal analysis, Funding acquisition, Methodology; Dominique Soldati-Favre, Conceptualization, Data curation, Formal analysis, Supervision, Funding acquisition, Validation, Project administration, Writing - review and editing

### Author ORCIDs

Nicolas Dos Santos Pacheco (iD) https://orcid.org/0000-0003-1959-194X
Paul Guichard (iD) https://orcid.org/0000-0002-0363-1049
Dominique Soldati-Favre (iD) https://orcid.org/0000-0003-4156-2109

### Decision letter and Author response

Decision letter https://doi.org/10.7554/eLife.56635.sa1
Author response https://doi.org/10.7554/eLife.56635.sa2

## Additional files

### Supplementary files

• Supplementary file 1. Top hits of AC9 BioID identified by mass spec with accession numbers, transcriptomic data, number of peptides detected and CRISPR-Cas9 essentiality score.

• Supplementary file 2. Mass spectrometry data of AC9-BirA experiment.

• Supplementary file 3. Key resources table.

- Supplementary file 4. List of primers used in this study.

- Transparent reporting form

## Data availability

All data generated or analysed during this study are included in the manuscript and Supplementary file 1.

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
