## [Decision Letter]

**Acceptance summary:**

Apicomplexan parasites possess a triple membrane pellicle made of the plasma membrane and inner membrane sacs called alveoli, which are intimately associated with subcortical microtubules. In addition, coccidian parasites including Toxoplasma harbor a unique apical structure called the conoid. Together with apical secretory organelles, these structures are essential for parasite motility, host cell invasion and egress. Here, using sophisticated high resolution microscopy and conditional reverse genetics, Tosetti et al. identify two key components of the parasite apical alveolin network that play an essential role in the assembly of subpellicular microtubules and conoid in *Toxoplasma gondii*. This study provides a substantial advancement to our understanding of the complex architecture of apical structures in apicomplexan parasites.

**Decision letter after peer review:**

Thank you for submitting your article "Essential function of the alveolin network in the subpellicular microtubules and conoid assembly in *Toxoplasma gondii*" for consideration by *eLife*. Your article has been reviewed by three peer reviewers, and the evaluation has been overseen by a Reviewing Editor and Anna Akhmanova as the Senior Editor. The following individual involved in review of your submission has agreed to reveal their identity: Rita Tewari (Reviewer #3).

The reviewers have discussed the reviews with one another and the Reviewing Editor has drafted this decision to help you prepare a revised submission.

Summary:

This is a very elegant study describing the role of novel alveolin proteins in assembly of the cytoskeleton of Toxoplasma. Coccidian parasites including Toxoplasma possess a unique structure called the conoid as part of the apical complex and used by the parasite for invasion, motility and egress. The authors capitalize on the latest techniques (BioID, super-resolution imaging techniques, including for the first time in Toxoplasma ultrastructure expansion microscopy, and conditional knockdown using the auxin degron system), to establish the function and in-depth localization of two alveolin proteins, AC9 and newly identified AC10, in assembly of the sub-cortical microtubules and conoid. Conditional depletion of AC9 or AC10 reveals defects in microneme secretion, egress and host cell invasion, associated with a loss of the apical polar ring and conoid and loss of integrity of the apical complex in mature parasites. This study provides a substantial advancement to our understanding of the complex architecture of apical structures in Toxoplasma.

All reviewers agree that this is a very elegant study and that no additional experimental work is required. However, they raised a number of comments that could be addressed or discussed to improve or clarify the manuscript.

Essential revisions:

The statistic in Figure 7 appears to be incorrect. Please revise.

Reviewer #1:

In this study, Tosetti and colleagues report on the role of two apical cap proteins, AC9 and AC10, in the stability of the apical complex in *Toxoplasma gondii*.

The work capitalizes on BioID (to identify AC10 as a partner for AC9) and super-resolution imaging techniques (STED, FIB-SEM, and for the first time in Toxoplasma ultrastructure expansion microscopy), combined with conditional knockdown using the auxin degron system. Conditional depletion of AC9 or AC10 reveals defects in microneme secretion, egress and host cell invasion, associated with a loss of the apical polar ring and conoid and loss of integrity of the apical complex in mature parasites. Interestingly, AC9 and AC10 deficient parasites still replicate normally, and the loss of the APR and conoid is observed in mature parasites but not in daughter cells.

This is a very elegant study that capitalizes on high-end tools and provides new insights into the assembly and maintenance of the apical complex in Toxoplasma. The work is very well performed and the data convincing. However, a few points could be clarified in the manuscript.

The authors propose that AC9 and AC10 act as a glue between subpellicular microtubules, a prerequisite to maintain the overall architecture of the key components of the apical complex when the parasite mature. However, how do they explain that the loss of conoid and APR only occur at the end of replication and not in the daughter cells? Is it possible that distinct mechanical forces during cytokinesis, applied on fragilized apical structures, are responsible for the observed phenomenon?

The authors show that a short treatment with IAA is sufficient to completely degrade AC9 and AC10 in AID parasites. I find this result difficult to reconcile with a proposed role of AC9 and AC10 as structural components acting as a glue between microtubules (I would expect such proteins to be resistant to AID degradation).

In Figure 6, the authors show that IAA treatment of extracellular mature parasites, although leading to degradation of AC9/AC10, preserves the conoid integrity and microneme secretion. What about motility and host cell invasion, and microtubule arrangement?

In the BioID experiments, the control consisted in BirA-tagged parasites without biotin. What happens with WT parasites in the presence of biotin?

Reviewer #2:

This is a well written and beautifully illustrated manuscript describing the role of several novel articulin proteins in assembly of the cytoskeleton of Toxoplasma. The authors employ the latest techniques to establish the role of AC9 and newly identified AC10 in assembly of the sub-cortical microtubules and novel apical MT organizing complex called the conoid in this parasite. Novel aspects include the use of super resolution, ultrastructure expansion microscopy and FIB-SEM. Together these techniques provide a substantial advancement to our understanding of the complex architecture of the conoid and subpellicular MTs.

I do not feel that additional data are necessary to support the conclusions. However, some sentences are awkwardly worded and could be improved in revision.

Reviewer #3:

This is a tour de force manuscript in analysing the function and in depth localisation of two essential alveolin proteins in conoid assembly. The coccidian parasites of Apicomplexa group possess conoid made of unique tubulin polymer fibres as part of apical complex and used by parasite for invasion, motility and egress. The authors have demonstrated for first time the use of U.ExM in any Apicomplexan parasite in precisely studying the apical polarity and conoid assembly of the Toxoplasma parasites. They have shown the function of two protein named Apical Protein AC9 and 10 and how they are involved in conoid assembly both in intracellular and extracellular parasite. The results are presented in very elegant and precise manner with proper controls and clear biochemical and statistical data. It sets the standards for this type of studies for cell biologist especially studying apicomplexan parasites.

My few comments are below which is not for any new experiment but some insight or discussion may help:

1) The one question that was intriguing is that these two proteins were affecting the conoid assembly in mature parasite but the daughter cells were not affected. So is the differentiation of conoid assembly a late event and maybe authors can speculate how this signalling may take place. It is fully appreciated that no experiments are needed but some implication that how centrosome division is linked to the conoid assembly will be useful. It is very interesting that conoid and microtubules in SPMT are affected in the absence of AC9. Is the SPMT is first event or the conoid assembly the first step in conoid formation.

2)The authors did mention that the absence of AC10 affects AC9 but not the other way round. So in the temporal profile present if I have not missed that it was not clear if the AC10 came before AC9 in the conoid localisation.

3) The authors have shown the centrin staining in the knockdown parasite. It was not clear what was the position of centrosome in the daughter parasite and how does the microtubule staining observed in the daughter parasites in the absence of AC9 or AC10.

4) Can author please implicate why daughter parasite conoid are not affected in the absence of AC9 or AC10. Is it possible that the auxin treatment is not able affect daughter parasites. May be too far fetched from me.

5) Recent studies by Leung et al., 2020, have also shown the role of double corti protein in tubulin assembly during conoid assembly. Maybe the authors can discuss their finding in relation to their observation in the Discussion.

For me the most open question why the daughter parasites conoid do not get affected in the absence of AC9 and AC10 and they almost behave like extracellular parasites. Of course the work for future studies.

I really enjoyed reviewing this manuscript and hope U.ExM presented by authors will help in many future cell biology studies to dissect out the cell polarity in Apicomplexan.

---

## [Author Response]

Essential revisions:The statistic in Figure 7 appears to be incorrect. Please revise.

The statistical analysis of the data presented in the Figure 7D and E have been revised and corrected.

For the Figure 7D, several tests were performed to be sure that the observed differences are significant. To be precise, Chi2 tests, standardized deviation from expected values and Fisher’s test were performed and all showed significant differences between the control group and the two mutants for each condition. In addition, to ensure the strength of our statistical analysis, the Cohen’s D coefficient was calculated for each condition and showed that we can have great confidence in our statistical analysis. The detail of the performed tests can be found in Figure 7—source data 1.

For the Figure 7E, unpaired T-test were performed as requested. To note, the differences observed are statistically significant between WT.vs.AC9+IAA and WT.vs.AC10+IAA. The detail of the performed tests can be found in Figure 7—source data 2. The figure has been modified accordingly. Also, there was a switch between AC9 and AC10 data in the original submission. This error has been corrected in the revised manuscript.

Reviewer #1:In this study, Tosetti and colleagues report on the role of two apical cap proteins, AC9 and AC10, in the stability of the apical complex in Toxoplasma gondii.The work capitalizes on BioID (to identify AC10 as a partner for AC9) and super-resolution imaging techniques (STED, FIB-SEM, and for the first time in Toxoplasma ultrastructure expansion microscopy), combined with conditional knockdown using the auxin degron system. Conditional depletion of AC9 or AC10 reveals defects in microneme secretion, egress and host cell invasion, associated with a loss of the apical polar ring and conoid and loss of integrity of the apical complex in mature parasites. Interestingly, AC9 and AC10 deficient parasites still replicate normally, and the loss of the APR and conoid is observed in mature parasites but not in daughter cells.This is a very elegant study that capitalizes on high-end tools and provides new insights into the assembly and maintenance of the apical complex in Toxoplasma. The work is very well performed and the data convincing. However, a few points could be clarified in the manuscript.The authors propose that AC9 and AC10 act as a glue between subpellicular microtubules, a prerequisite to maintain the overall architecture of the key components of the apical complex when the parasite mature. However, how do they explain that the loss of conoid and APR only occur at the end of replication and not in the daughter cells? Is it possible that distinct mechanical forces during cytokinesis, applied on fragilized apical structures, are responsible for the observed phenomenon?

This is indeed an intriguing and central question regarding the interpretation of the AC9 and AC10 phenotypes. In contrast to the mature mother parasite that faces the external milieu, the daughter cells in the cytosol of the mother cells and hence the mechanical forces applied to the growing SPMTs cytoskeleton is presumably reduced compared to tension imposed on the SPMTs of the mature parasite in order to maintain the shape and cope with the osmotic pressure.

In our view the most plausible, possibly naïve, explanation, is that at the time when the daughter cells emerge from the mother cell, the tension imposed on the SPMTs leads to their disassembly and consequently to the loss of APR and conoid. We are presenting this speculative model more explicitly in the Discussion section.

The authors show that a short treatment with IAA is sufficient to completely degrade AC9 and AC10 in AID parasites. I find this result difficult to reconcile with a proposed role of AC9 and AC10 as structural components acting as a glue between microtubules (I would expect such proteins to be resistant to AID degradation).

This might appear counterintuitive although several parameters influence the kinetics of degradation of a given mAID-tagged protein. The intrinsic half-life (turnover) of a protein is not easily predictable and many structural proteins are known to be very dynamics. The AiD system is vastly accelerating this turnover. The subcellular localization of a given protein and its access to proteasome is also likely to significantly influence the speed of degradation. Among the many proteins that we have studied via mAID degron system in the laboratory, we have been astonished to observe overall very rapid kinetics of degradation.

In Figure 6, the authors show that IAA treatment of extracellular mature parasites, although leading to degradation of AC9/AC10, preserves the conoid integrity and microneme secretion. What about motility and host cell invasion, and microtubule arrangement?

These phenotypic experiments were not performed. Noteworthy either wild type or AC9-mAID parasites maintained extracellularly for 2 to 8 hours are anticipated to gradually and considerably lose their ability to glide and to invade host cells. The presence of the conoid and all the apical proteins as well as the ability to secrete of microneme presumably indicates that the parasites are motile and invasive. However, given the drop in microneme secretion we anticipate a significant decrease in motility and invasion both in presence and in absence of IAA! Regarding the SPTMs arrangement, it turns out to be too challenging to treat extracellular parasites for 8h while keeping them fresh enough to perform U-ExM.

These complementary data would be valuable but not essential to the conclusions. Given the exceptional circumstance of confinement imposed by Covid-19 pandemia, they could not be performed for the revision but are discussed in the text.

In the BioID experiments, the control consisted in BirA-tagged parasites without biotin. What happens with WT parasites in the presence of biotin?

When performing the AC9-BirA experiment, we did not include WT parasites in the presence of biotin as control. However, we have performed such controls on several occasions in previous BioID experiments and such controls have been published by others when establishing the methodology for *Toxoplasma gondii.* In WT parasites there is endogenous biotinylation of the apicoplast just by looking at streptavidin staining without adding exogenous biotin.

Please find below the results of such a control experiment with WT RH and unrelated BirA tagged proteins (APH and AMA1)

Author response table 1

Reviewer #3:This is a tour de force manuscript in analysing the function and in depth localisation of two essential alveolin proteins in conoid assembly. The coccidian parasites of Apicomplexa group possess conoid made of unique tubulin polymer fibres as part of apical complex and used by parasite for invasion, motility and egress. The authors have demonstrated for first time the use of U.ExM in any Apicomplexan parasite in precisely studying the apical polarity and conoid assembly of the Toxoplasma parasites. They have shown the function of two protein named Apical Protein AC9 and 10 and how they are involved in conoid assembly both in intracellular and extracellular parasite. The results are presented in very elegant and precise manner with proper controls and clear biochemical and statistical data. It sets the standards for this type of studies for cell biologist especially studying apicomplexan parasites.My few comments are below which is not for any new experiment but some insight or discussion may help:1) The one question that was intriguing is that these two proteins were affecting the conoid assembly in mature parasite but the daughter cells were not affected. So is the differentiation of conoid assembly a late event and maybe the authors can speculate how this signalling may take place. It is fully appreciated that no experiments are needed but some implication that how centrosome division is linked to the conoid assembly will be useful. It is very interesting that conoid and microtubules in SPMT are affected in the absence of AC9. Is the SPMT is first event or the conoid assembly the first step in conoid formation.

The conoid appears early during daughter cell formation. The conoid as well as the SPMTs are formed normally in the daughter cells of AC9 and AC10 depleted parasites (see Figure 7C middle panels). We suspect that the SPMTs need to be tightly associated to the alveolin network to maintain the apical polar in place (APR) in place. In the absence of AC9 or AC10 the SPTMs are disassembled when the daughter cells are merging from the mother cells. As mentioned in response to the first point of reviewer #1.

In our view the most plausible explanation is that at the time when the daughter cells emerge from the mother cell, the tension imposed on the SPMTs leads to their disassembly and consequently to the loss of APR and conoid.

2)The authors did mention that the absence of AC10 affects AC9 but not the other way round. So in the temporal profile present if I have not missed that it was not clear if the AC10 came before AC9 in the conoid localisation.

The two proteins are both recruited at the earliest time during daughter cell formation. We postulate that AC10 is associated to the IMC whereas AC9 binds to AC10 and is stabilized as an AC9/10 complex. Concordantly, AC9 is lost in the absence of AC10 and not vice versa. This does not imply that the two proteins appear at different time. In any case the timing of recruitment might be too close to one another to be resolved experimentally.

3) The authors have shown the centrin staining in the knockdown parasite. It was not clear what was the position of centrosome in the daughter parasite and how does the microtubule staining observed in the daughter parasites in the absence of AC9 or AC10.

The centrosomes and annuli (strained by centrin2) are not affected in the AC9 or AC10 depleted parasites. Only the preconoidal ring staining of centrin2 is affected.

4) Can author please implicate why daughter parasite conoid are not affected in the absence of AC9 or AC10. Is it possible that the auxin treatment is not able affect daughter parasites. Maybe too far fetched from me.

See the explanation provided for point 2 of this reviewer.

We know that auxin treatment is perfectly capable of affecting daughter cell parasites. These data are derived from daughter cell-specific DHHC proteins localized only to nascent daughter cells and impacting on IMC formation (Frenal, unpublished).

5) Recent studies by Leung et al., 2020, have also shown the role of double cortin protein in tubulin assembly during conoid assembly. Maybe the authors can discuss their finding in relation to their observation in the Discussion.

The work from Leung et al., 2020, is now published so we have updated the reference list accordingly.

TgDCX is a protein associated with the conoid fibers, which plays a crucial role in curving the tubulin-polymers to build up a sable conoid and support efficient host-cell invasion. We have added comments on these findings in the Discussion section.